# Optimal Humanized Scg3-Neutralizing Antibodies for Anti-Angiogenic Therapy of Diabetic Retinopathy

**DOI:** 10.3390/ijms25179507

**Published:** 2024-09-01

**Authors:** Chengchi Huang, Prabuddha Waduge, Avinash Kaur, Hong Tian, Christina Y. Weng, John Timothy Stout, Iok-Hou Pang, Keith A. Webster, Wei Li

**Affiliations:** 1Cullen Eye Institute, Department of Ophthalmology, Baylor College of Medicine, Houston, TX 77030, USA; 2Everglades Biopharma, LLC, Houston, TX 77098, USA; 3Department of Pharmaceutical Sciences, North Texas Eye Research Institute, University of North Texas, Fort Worth, TX 76107, USA; 4Department of Pharmacology, Vascular Biology Institute, University of Miami Miller School of Medicine, Miami, FL 33136, USA

**Keywords:** secretogranin III, Scg3, anti-Scg3 therapy, Scg3-neutralizing antibodies, diabetic retinopathy, choroidal neovascularization

## Abstract

Secretogranin III (Scg3) is a diabetic retinopathy (DR)-restricted angiogenic factor identified in preclinical studies as a target for DR therapy. Previously, our group generated and characterized ML49.3, an anti-Scg3 monoclonal antibody (mAb) which we then converted into an EBP2 humanized antibody Fab fragment (hFab) with potential for clinical application. We also generated anti-Scg3 mT4 mAb and related EBP3 hFab. In this study, to identify the preferred hFab for DR therapy, we compared all four antibodies for binding, neutralizing and therapeutic activities in vitro and in vivo. Octet binding kinetics analyses revealed that ML49.3 mAb, EBP2 hFab, mT4 mAb and EBP3 hFab have Scg3-binding affinities of 35, 8.7, 0.859 and 0.116 nM, respectively. Both anti-Scg3 EBP2 and EBP3 hFabs significantly inhibited Scg3-induced proliferation and migration of human umbilical vein endothelial cells in vitro, and alleviated DR vascular leakage and choroidal neovascularization with high efficacy. Paired assays in DR mice revealed that intravitreally injected EBP3 hFab is 26.4% and 10.3% more effective than EBP2 hFab and aflibercept, respectively, for ameliorating DR leakage. In conclusion, this study confirms the markedly improved binding affinities of hFabs compared to mAbs and further identifies EBP3 hFab as the preferred antibody to develop for anti-Scg3 therapy.

## 1. Introduction

Anti-angiogenic drugs targeting vascular endothelial growth factor (VEGF), approved by the U.S. Food and Drug Administration (FDA), represent a major breakthrough in the therapy of diabetic retinopathy (DR) [1]. VEGF inhibitors are also the cornerstone of approved drug therapies for wet age-related macular degeneration (AMD) with choroidal neovascularization (CNV) [2]. Despite such a breakthrough, anti-VEGF therapy has multiple limitations, including suboptimal efficacy in certain patients, indiscriminate inhibition of both physiological and pathological angiogenesis and the bystander suppression of VEGF’s neurotrophic and neuroprotective activities [3,4,5,6]. An important strategy to improve the treatment efficacy and safety is through alternative or combination therapies that target other VEGF-independent angiogenic factors or signaling pathways [7]. Such a strategy has been explored, but with limited success. Fovista and nesvacumab, drugs that target platelet-derived growth factor and angiopoietin-2, respectively, failed in combination anti-VEGF clinical trials for wet AMD [8,9], probably because of common VEGF-dependent mechanisms of action [10,11,12]. Despite the apparent failure of nesvacumab, faricimab (Vabysmo) targeting VEGF and angiopoietin-2 was approved for wet AMD and DR based on non-inferiority clinical trials [13,14]. A synergistic combination anti-angiogenic therapy is yet to be developed.

We recently reported secretogranin III (Scg3) as a disease-restricted VEGF-independent angiogenic factor that selectively drives angiogenesis of diseased but not healthy vessels [15,16]. We generated ML49.3, an Scg3-neutralizing monoclonal antibody (mAb), and its derivative EBP2 humanized antibody (hAb) and demonstrated alleviation of DR leakage and CNV in animal models with efficacy equivalent to that of the anti-VEGF drug aflibercept [15,16,17,18]. Furthermore, the combination of anti-Scg3 hAb Fab fragment (hFab) with aflibercept synergistically ameliorated DR leakage and CNV [17,18]. During the development of anti-Scg3 therapy, one of the technical challenges was determining how to reliably compare therapeutic efficacy of different anti-Scg3 antibody clones in animal models. 

To improve treatment efficacy, we generated a second anti-Scg3 mT4 mAb from which EBP3 hFab was derived. The objective of this study is to compare these two sets of antibodies, ML49.3 mAb/EBP2 hFab and mT4 mAb/EBP3 hFab, for binding affinity, neutralizing activity and therapeutic efficacy in alleviating DR leakage and CNV in animal models. An additional objective is to assess the reliability of the DR mouse model for comparing therapeutic agents. 

## 2. Results

### 2.1. Binding Activity

The binding of ML49.3 and mT4 mAbs to human Scg3 (hScg3) has been previously reported [15,19]. Here, we compared the binding of full-length EBP2 and EBP3 hAbs to immobilized Scg3 by ELISA and confirmed similar Scg3-binding activities to hScg3 and mouse Scg3 (mScg3) (Figure 1A). 

Previously, our group reported binding affinities (K_D_) of 35 and 8.7 nM for ML49.3 mAb and EBP2 hFab, respectively, using an Octet instrument [17]. In this study, employing the same instrument, we determined the K_D_ values for mT4 mAb and EBP3 hFab to be 0.859 and 0.116 nM, respectively (Figure 1B,C). The results indicate that EBP3 has the highest binding affinity (Table 1).

### 2.2. Detection of Scg3 by Immunohistochemistry

Immunohistochemistry was employed to detect Scg3 expression in the retina using all four antibodies. Both mT4 mAb and EBP3 hAb revealed Scg3 expression predominantly in the retinal ganglion cell layer (GCL), inner plexiform layer (IPL), outer plexiform layer (OPL) and photoreceptor inner segments (PIS), but not in the inner and outer nuclear layers (INLs and ONLs) (Figure 2, Rows 5 and 2, respectively). Scg3 expression was not detected in the retina of Scg3^−/−^ mice using EBP3 hAb. This pattern is consistent with Scg3 expression in neurotransmitter vesicles [20] and with the expression profile previously detected using anti-Scg3 polyclonal antibodies [15,16,21]. In contrast, neither ML49.3 mAb nor EBP2 hAb immunostaining revealed significant signals above background (Figure 2, Rows 6 and 3, respectively). Notably, the bright signals in the choroidal layer visualized by mT4 and ML49.3 mAbs (Figure 2, Rows 5 and 6, respectively) are likely due to residual endogenous mouse IgG following intracardial perfusion by anti-mouse IgG secondary antibody. The anti-mouse IgG secondary antibody alone also revealed similar non-specific signals in the mouse choroids (Figure 2, Row 4).

### 2.3. Neutralizing Activity

As shown in Figure 3A, EBP2 and EBP3 hFabs abolished Scg3-augmented proliferation of human umbilical vein endothelial cells (HUVECs), with no significant difference in neutralization activity between the two hFabs. As a positive control, VEGF-induced endothelial proliferation was significantly blocked by aflibercept. 

We further characterized the neutralizing activity of EBP2 and EBP3 hFabs by evaluating their ability to block Scg3-induced transwell migration of HUVECs. Both hFabs significantly inhibited HUVEC migration to a similar degree (Figure 3B,C). As a positive control, aflibercept suppressed VEGF-induced HUVEC migration.

### 2.4. Alleviation of CNV

We previously reported the inhibition of CNV in animal models by anti-Scg3 ML49.3 mAb and EBP2 hFab [16,17]. Here, we quantified laser-induced CNV in the presence or absence of anti-Scg3 mT4 mAb. As shown in Figure 4A–C, CNV leakage area and intensity quantified by fluorescein angiography were significantly ameliorated by mT4 mAb. Similarly, immunostaining of flat-mount retinal pigment epithelium (RPE)–choroid–sclera complexes (i.e., RPE eyecups) by Alexa fluor 488-isolectin B4 (AF488-IB4) revealed that mT4 mAb markedly inhibited the lesion size and 3D volume of CNV vessels (Figure 4D–F). 

In parallel experiments, we further characterized the therapeutic activity of the related EBP3 hFab in ameliorating laser-induced CNV in mice (Figure 5). The results showed that EBP3 hFab also significantly alleviated CNV leakage size and intensity (Figure 5A,C,D), and immunostaining of CNV vessels in RPE eyecups confirmed a similar reduction in CNV lesion size and 3D volume by EBP3 hFab (Figure 5B,E,F).

### 2.5. Amelioration of DR

We previously reported reduced DR leakage in diabetic retinopathy by treatment with ML49.3 mAb and EBP2 hFab, including dose-dependence of EBP2 hAb [15,18]. Here, we also confirmed significant inhibition of DR leakage by anti-Scg3 mT4 mAb (Figure 6A,B), and high therapeutic efficacy of EBP3 hFab which dose-dependently alleviated DR leakage in diabetic mice (Figure 6C).

### 2.6. Efficacy Comparison

Because of the high inherent variability of hyperglycemia and subsequent DR severity in individual mice, we compared the therapeutic efficacies of EBP2 and EBP3 hFabs between the left and right eyes of each DR mouse by intravitreally injecting equal doses of the blind-coded paired antibodies, as described in the Methods section. DR leakage in both eyes was quantified by Evans blue assay. After data normalization against the blood level of Evans blue and retinal weight, the DR leakage index was decoded, normalized to the EBP2-treated eye, and analyzed using a paired t-test to compare EBP2 and EBP3 by pairing the two eyes of each mouse. By this method, we found that EBP3 is 24.6% more effective than EBP2 (Figure 6D, *p* < 0.0001). Interestingly, a regular two-tailed t-test analysis of the same dataset showed no statistical significance (Figure 6D, *p* = 0.146).

In previous studies, we found that EBP2 hFab and aflibercept were therapeutically equivalent based on unpaired regular statistical analyses [17,18,21]. When we applied the same paired experimental design described in Figure 6D to compare EBP3 and aflibercept, the same paired t-test analysis revealed that EBP3 hFab blocked DR leakage 10.3% more effectively than aflibercept (Figure 6E, *p* < 0.05). Once again, regular *t*-test comparison of the same dataset revealed no statistical difference (Figure 6E, *p* = 0.601).

## 3. Discussion

Scg3 is a disease-restricted angiogenic factor that selectively stimulates angiogenesis and vascular leakage of diseased but not healthy vessels [15,16,18,22]. We previously reported the advantage of anti-Scg3 ML49.3 mAb and EBP2 hAb, which stringently inhibits pathological but not physiological angiogenesis in animal models of retinopathy of prematurity (ROP) [21,23]. Dose–response curves in mouse models of DR, CNV and ROP also revealed that EBP2 hFab and aflibercept have similar therapeutic efficacies [17,18,21]. Here, by using comprehensive binding and functional and therapeutic efficacy assays, we show that both hFabs have markedly improved antigen binding compared with their respective cognate mAbs. 

Both ML49.3 mAb and EBP2 hFab recognize mScg3 and hScg3, block hScg3-induced proliferation of endothelial cell in vitro and alleviate pathological angiogenesis and vascular leakage in mouse models of DR, CNV and ROP [15,16,17,18,21,23,24]. Therefore, we attribute the failure of ML49.3 and EBP2 hFab to detect mouse Scg3 in immunohistochemistry (Figure 2) to the recognition of a conformational epitope that is disturbed by tissue fixation, unrelated to species-specific antigen recognition. It is also noteworthy that the markedly increased binding affinity of EBP3 compared to EBP2 did not translate proportionally into neutralizing activity and therapeutic efficacy (Table 1 vs. Figure 3 vs. Figure 6B–D). This may also be attributed to differences in epitope presentation, binding affinities, antibody conformation and penetration in the vivo milieu versus that of the in vitro assay. As discussed further below, the results support paired in vivo efficacy assays for comparing anti-angiogenic biologic therapies.

Due to the high inherent variability of murine models of diabetes, including degrees of hyperglycemia and resulting disease severity, it can be technically challenging to quantitatively compare the preclinical efficacies of different drugs. As a result, excessively large sample sizes are often required to detect statistical significance.

Here, we designed a paired DR assay to compare the therapeutic efficacies of EBP2 and EBP3 hFab using the following approach (Figure 6D). Equal doses of the two therapeutic agents were paired, blind-coded and intravitreally injected into randomly assigned left and right eyes of recipient DR mice. Although individual diabetic mice exhibit varying severities of hyperglycemia and DR responses, paired eyes of the same mice typically experience equivalent degrees of DR leakage severity due to shared systemic hyperglycemia and similar histopathological DR responsiveness. Additionally, during the Evans blue assay, both eyes are intracardially perfused in parallel, which eliminates most experimental variations. Data variation may still occur due to in vitro Evans blue assay procedures, such as retinal dissection and dye extraction, but these procedures are also standardized to minimize variation. As a result, the paired DR leakage assay reliably demonstrated improved anti-DR therapy by EBP3 relative to EBP2 hFab and aflibercept using as few as five individual mice with statistical significance demonstrated by paired *t*-test (Figure 6D,E). In contrast, a regular two-tailed t-test of the same datasets revealed no statistical difference due to variations in disease severity among individual mice.

The paired DR assay was also used to quantify therapeutic efficacy and potency of individual anti-angiogenic/leakage agents by pairing with PBS or control IgG. The dose–response curve shown in Figure 6C was generated by pairing EBP3 hFab with PBS in five sets of DR mice (n = 5 mice/group, one dose/group), although only one set of the representative PBS control group is shown (Figure 6C, left-most column). All five sets of the therapeutic efficacies were normalized to cognate paired PBS eyes of the same DR mice to reduce variations among individual DR mice before plotting the dose–response curve (Figure 6C). The same paired approach was also used in a previous study to quantify the synergistic combination therapy of anti-Scg3 hAb and aflibercept against monotherapies to ameliorate DR leakage [18]. We recommend using such an approach in the DR mouse model to more accurately define and compare therapeutic efficacies relative to other animal models, such as laser-induced CNV, where inter-mouse variability can necessitate large sample sizes for reliable efficacy comparison.

This study characterized Scg3 as an angiogenic factor in a CNV model and a vascular leakage factor in a DR model [15]. Many angiogenic factors, such as VEGF and IL-8, also promote vascular leakage [25,26,27], whereas other angiogenic factors, including angiopoietin-1, can stabilize vascular integrity and suppress vascular leakage [28]. Some vascular leakage factors, such as semaphorin 3A, are also capable of inhibiting angiogenesis [29]. Retinal vascular leakage is an early clinical manifestation in DR patients that may advance to proliferative diabetic retinopathy (PDR) with pathological retinal neovascularization 15–20 years after the onset of diabetes [30]. In contrast, diabetic mice usually develop retinal vascular leakage but not PDR, probably because of their relatively short lifespans. Therefore, the paired comparison described here for DR mice is broadly applicable to different vascular leakage and angiogenic factors after validating their pathogenic roles and therapeutic potentials.

This study used the DR mouse model to reliably compare the therapeutic efficacy of anti-Scg3 antibodies and aflibercept, selecting EBP3 hFab as an optimal candidate for clinicial translation. Significant obstacles remain for clinical translation, including Good Manufacturing Practice (GMP) antibody production, Good Laboratory Practice (GLP) toxicology and demonstration of synergistic combinations with anti-VEGF in clinical trials.

In summary, this study compared anti-Scg3 ML49.3, EBP2, mT4, and EBP3 mAbs and hFabs for binding, neutralizing and therapeutic activity in DR mice with emphasis on the hFabs. We conclude that EBP3 hFab has superior binding affinity compared to EBP2 hFab and alleviates DR leakage in diabetic mice more effectively than either EBP2 or aflibercept. The novel paired preclinical assay quantitatively, reliably and efficiently revealed the efficacies of EBP2 vs. EBP3 hFab in alleviating retinal vascular leakage in DR mice. This assay can be broadly applied for preclinical anti-angiogenic drug screening to select the best candidates for clinical translation. Based on this work, EBP3 hFab has been selected for Investigational New Drug (IND)-enabling preclinical studies to advance the biologic toward clinical trials.

## 4. Materials and Methods

### 4.1. Materials

Scg3-neutralizing ML49.3 mAb and EBP2 hFab were previously reported [15,17]. Anti-Scg3 mT4 mAb and the related EBP3 hFab were generated by Everglades Biopharma, LLC [19]. C57BL/6J mice were purchased from the Jackson Laboratory (Bar Harbor, ME, USA), and Scg3-knockout mice were obtained from Taconic Biosciences (Rensselaer, NY, USA) [17].

### 4.2. ELISA

Human Scg3 (Sino Biological, Wayne, PA, USA; Cat. #16012-H08H) or mouse Scg3 (Sino Biological, Cat. # 51561-M08H) was immobilized on ELISA plates, blocked, and incubated with anti-Scg3 EBP2 and EBP3 hAb [15]. Bound hAbs were detected using horseradish peroxidase-conjugated goat anti-human IgG (Sigma, St. Louis, MO, USA, Cat. #A0293), followed by a colorimetric assay, as described [15,31].

### 4.3. Binding Kinetics Measurements

Binding kinetics measurements were performed using an Octet QKe system (ForteBio, Fremont, CA, USA), as described [17,32]. Briefly, human Scg3 (hScg3) was labeled with biotin using NHS-PEG4-Biotin (Thermo Fisher Scientific, Waltham, MA, USA, Cat #A39259), followed by desalting purification. Biotin-Scg3 was loaded onto streptavidin biosensors in the Octet instrument, washed, and bound to increasing concentrations of purified anti-Scg3 mT4 mAb and EBP3 hFab. Antibody binding affinities were calculated using the Octet software (ForteBio, Data Analysis, version 11.1.0.4).

### 4.4. Endothelial Proliferation Assay

HUVECs (Lonza, Hopkinton, MA, USA) were maintained in Complete Classic Medium with serum and culture boost (Cell Systems, Cat. # 4Z0-500) and 1% penicillin/streptomycin (Hyclone, Logan, UT, USA) at 37 °C and 5% CO_2_ [33]. For endothelial proliferation assay, HUVECs were seeded in the Complete Classic Medium at 1 × 10^4^ cells/well in 96-well plates and cultured overnight. The media were replaced with EBM-2 medium (Lonza, Cat #CC-3156) supplemented with 2% FBS. hScg3 (1 µg/mL) or hVEGF (100 ng/mL, R&D Systems, Minneapolis, MN, USA, Cat. #BT-VEGF-50) was added to individual wells in the presence or absence of indicated blocking reagents (2 µg/mL) or phosphate-buffered saline (PBS). After 48 h, cell numbers were quantified.

### 4.5. Endothelial Transwell Migration Assay

The assay was performed in vitro as previously described [34]. In brief, HUVECs (5 × 10^4^ cells/well) were seeded into the upper chamber of transwell inserts in 24-well plates (8 μm pore size and 6.5 mm diameter, Corning Life Science, Corning, NY, USA, #3422) precoated with 1% gelatin. hScg3 (1 µg/mL) or hVEGF (100 ng/mL) was added to the lower chamber along with the indicated blocking reagents (2 µg/mL) or PBS. After 20 h in culture, the transwell inserts were removed and fixed with 4% paraformaldehyde for 10 min. Cells were wiped off from the upper side of the membrane with a cotton swab, stained with DAPI, mounted on slides, and analyzed by fluorescence microscopy to quantify migrated cells on the surface of the lower chamber.

### 4.6. Immunohistochemistry

Mice were euthanized by CO_2_ inhalation and immediately perfused intracardially with PBS (50 mL), followed by 4% paraformaldehyde (20 mL) and then eye enucleation. The anterior segments, including the cornea, iris, and lens, were removed to yield eyecups with the retina that were embedded in the optimal cutting temperature (OCT) compound (Tissue-Tek; Miles Scientific, Naperville, IL, USA) and cryosectioned in 10 μm thickness. Eye sections were immunostained with purified anti-Scg3 ML49.3, EBP2, mT4 or EBP3 antibodies, followed by Alexa Fluor 594-conjugated goat anti-mouse IgG antibody or Alexa Fluor 594-conjguated goat anti-human IgG antibody (Cell Signaling and Thermo Fisher, Waltham, MA, USA, Cat. #8890 and #A1101, respectively; dilution 1:1000 for both). Nuclei were visualized with Hoechst and analyzed using a Keyence structure illumination microscope (SIM, Model BZX-810).

### 4.7. DR Mice

C57BL/6J male mice (6–8 weeks of age) were treated intraperitoneally with streptozotocin (STZ, 50 mg/kg body weight, for 5 consecutive days; Sigma, Cat #S0130) in citrate buffer to induce type 1 diabetes [15]. Hyperglycemic mice (blood glucose > 300 mg/dL) were then aged for 4 months to develop chronic DR.

### 4.8. Evans Blue Leakage Assay

Retinal vascular permeability in DR mice was quantified using the Evans blue assay [15]. Briefly, therapeutic reagents at indicated concentrations were intravitreally injected into one eye of anesthetized mice with control reagent or PBS for the contralateral eyes of the same mice in a blind-coded manner. After 24 h, Evans blue (20 mg/mL) was administered intravenously. The mice were maintained on a heating pad for 4.5 h, followed by intracardiac perfusion with pre-warmed (37 °C) citrate buffer (100 mM, pH 3.5) for 30 min via the left ventricle to remove the circulating dye. Retinas were dissected from the enucleated eyes, weighed and incubated with 50 µL formamide per retina at 70 °C for 24 h to extract Evans blue. The solution was centrifuged at 180,000× *g*, for 1.25 h at 4 °C, and quantified at 620 and 740 nm (background). Blood samples were collected from dye-injected mice immediately before intracardial perfusion and centrifuged at 3550× *g* for 10 min at 25 °C. Serum was collected, diluted 1 to 50 into formamide and quantified for serum concentration of Evans blue. To assess the leaked blue, the absorbance of the retinal extract and plasma samples was compared to a standard curve. Evans blue leakage was calculated using the following equation: (retina leaked EB concentration [mg/mL]/retinal weight [mg])/(blood EB concentration [mg/mL] × circulation time [h]) [15]. Data are normalized to the control in the fellow eyes of the same mice and expressed as a percentage of reduction in leakage.

### 4.9. Paired Assay for Efficacy Comparison

To reliably compare the therapeutic efficacy, EBP2 hFab was paired with EBP3 hFab or aflibercept in equal doses, blind-coded and intravitreally injected individually into the left or right eye of each diabetic mouse. DR leakage in both eyes was quantified by the Evans blue assay as described above. After data normalization against the blood level of Evans blue and retinal weight, DR leakage index was decoded, normalized to the EBP2-treated eye of the same mouse and analyzed using a paired *t*-test for EBP2 against. EBP3 by pairing two eyes of each mouse.

### 4.10. Laser-Induced CNV Mouse Model

C57BL/6J mice (6–8 weeks old, male or female) were anesthetized by intraperitoneal injection of ketamine hydrochloride (80 mg/kg) and xylazine (16 mg/kg). The pupils were dilated using topical 1% tropicamide (Akorn, Lake Forest, IL, USA) and 2.5% phenylephrine (Paragon BiTeck, Portland, ME, USA) (one drop each). A 532 nm green laser with the Micron IV retinal imaging system (Phoenix Research Labs, Pleasanton, CA, USA) was used to burn the fundus of the mice (4 laser spots/eye) [19,24]. The burn spots were located at the 3, 6, 9 and 12 o’clock positions around the optic disc (spot size: 50 μm; duration: 140 ms; power: 240 mW). The formation of gaseous bubbles at laser spots indicated the rupture of Bruch’s membrane. Linear or merged lesions, as well as lesions with retinal bleeding were excluded.

### 4.11. Fluorescein Angiography

Tropicamide (1%) and phenylephrine (2.5%) were applied topically onto the cornea for pupillary dilation of anesthetized mice, as described above. CNV leakage was recorded 7 days post laser photocoagulation using fundus fluorescein angiography (FFA), which was performed with a Spectralis Tracking OCTA system (Heidelberg Engineering, Franklin, MA, USA). FFA images were captured 6 min after intraperitoneal injection of 100 μL of 2.5% sodium fluorescein (Akorn). The leakage area and mean intensity value of each lesion were quantified using ImageJ software (version # 1.54d; National Institutes of Health, Bethesda, MD).

### 4.12. CNV Vessel Staining

After fluorescein angiography, CNV mice were euthanized, and eyes were enucleated. Following the removal of the anterior sections and retina, RPE eyecups were isolated, stained with Alexa Fluor 488-isolectin B4 (Thermo Fisher, Cat. #I21411, 10 µg/mL) to label CNV vessels, which were imaged under the Keyence SIM and quantified using the related software [19,24].

### 4.13. Statistical Analysis

Data are expressed as mean ± SEM. Intergroup differences were analyzed using ANOVA with a Tukey post hoc test, a two-tailed Student’s *t*-test, or paired *t*-test.

## 5. Conclusions

We used DR mice as a model to compare two therapeutic agents with a negligible influence on disease severity in individual animals and determined that EBP3 hFab is significantly more effective than EBP2 hAb and aflibercept. The results suggest that anti-Scg3 EBP3 hFab is an optimal candidate for clinical translation. The in vivo comparison assay in the DR model is also valuable for reliably evaluating efficacy of any two anti-angiogenic biologics.

## Figures and Tables

**Figure 1 ijms-25-09507-f001:**
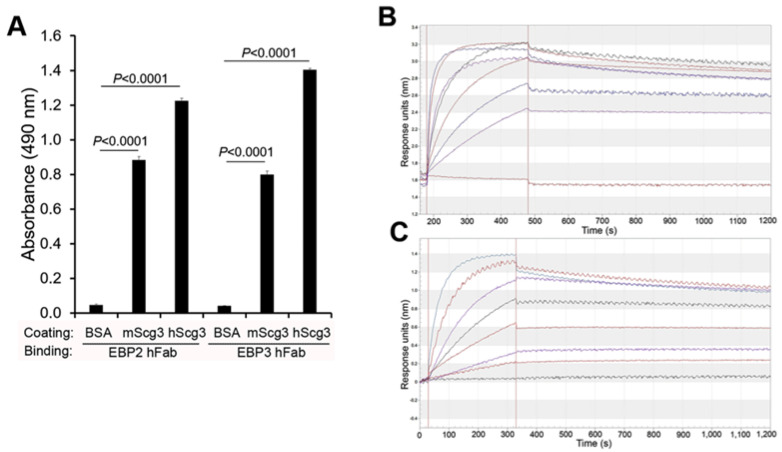
**Characterization of anti-Scg3 mAb and hFab.** (**A**) Anti-Scg3 EBP2 and EBP3 hFab bind to both mouse Scg3 (mScg3) and human Scg3 (hScg3) as demonstrated by ELISA. In this assay, mScg3, hScg3 and control bovine serum albumin (BSA) were immobilized on ELISA plates, which were then blocked, incubated with EBP2 and EBP3 hFab, washed, and detected by enzyme-conjugated goat anti-human IgG secondary antibody and a colorimetric assay. n = 3 well/group. ±SEM, one-way ANOVA test. (**B**,**C**) The binding kinetics of mT4 mAb (**B**) and EBP3 hFab (**C**) were analyzed using an Octet instrument to calculate binding affinity (see Table 1).

**Figure 2 ijms-25-09507-f002:**
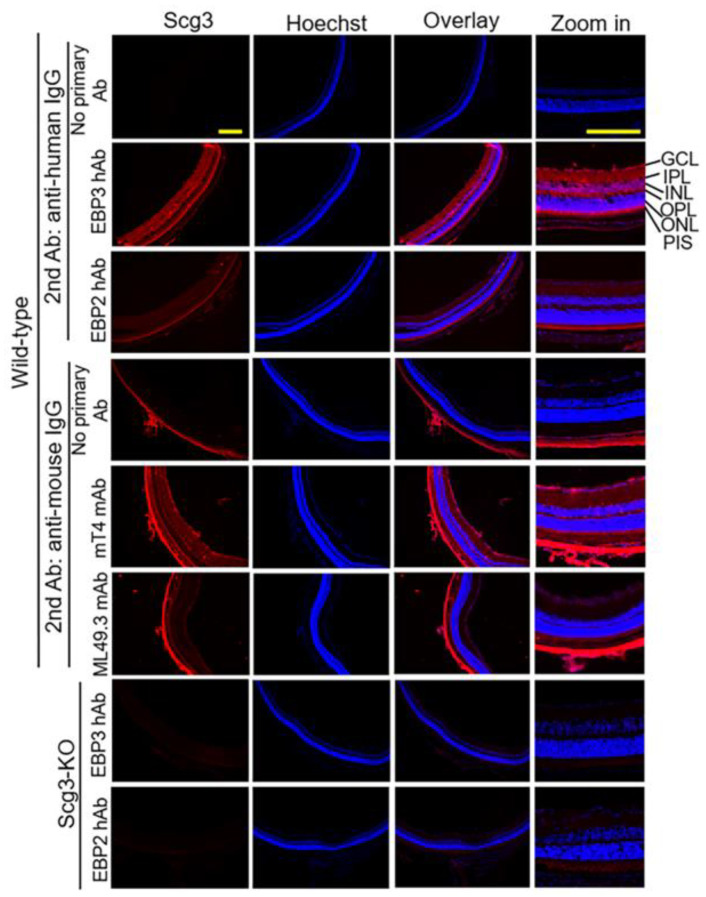
Immunohistochemistry was performed to detect the retinal expression of Scg3 in wild-type and Scg3^−/−^ mice using anti-Scg3 ML49.3, mT4, EBP2 and EBP3 antibodies. GCL—retinal ganglion cell layer; IPL—inner plexiform layer; INL—inner nuclear layer; OPL—outer plexiform layer; ONL—outer nuclear layer; PIS—photoreceptor inner segment. Scale = 200 μm.

**Figure 3 ijms-25-09507-f003:**
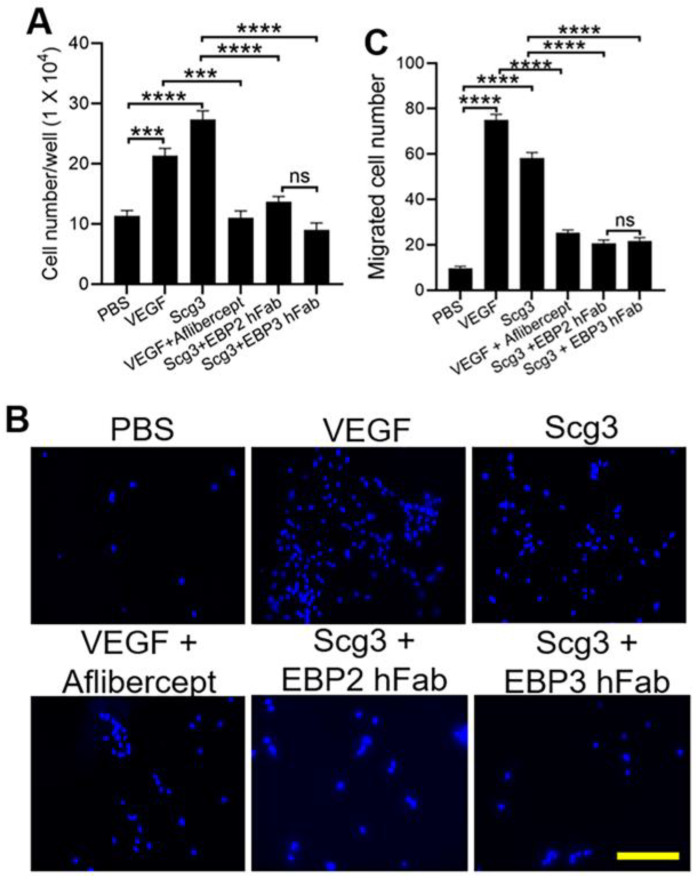
Neutralizing activity of anti-Scg3 EBP2 and EBP3 hFabs. (**A**) Both anti-Scg3 EBP2 and EBP3 hFabs inhibited Scg3-induced proliferation of HUVECs. Cells were treated with VEGF (100 ng/mL) or Scg3 (1 μg/mL) in the presence or absence of aflibercept (2 μg/mL) or anti-Scg3 hFab (2 μg/mL), respectively. (**B**) Representative images of migrated cells in the transwell migration assay. Cells were incubated in transwell inserts with indicated reagents at the same concentrations as described in A. Scale = 200 μm. (**C**) Quantification of transwell migrated cells. ±SEM; n = 3 wells/group; *** *p* < 0.001, **** *p* < 0.0001; ns—not significant; one-way ANOVA test.

**Figure 4 ijms-25-09507-f004:**
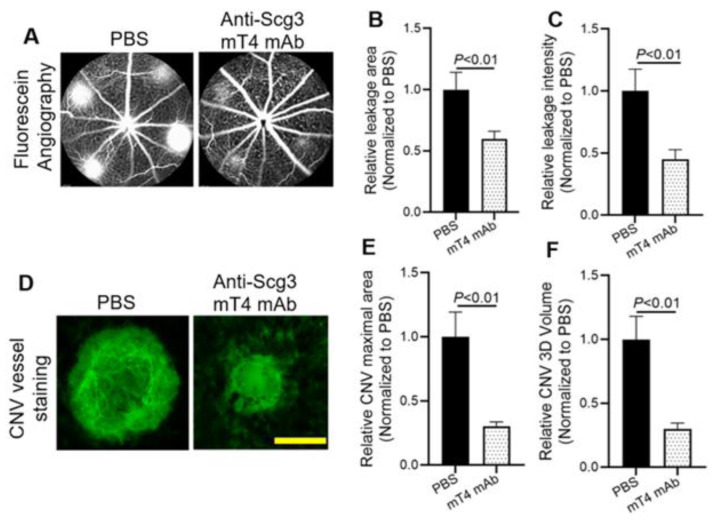
Anti-Scg3 mT4 mAb alleviates laser-induced CNV (LCNV). Mice were induced for LCNV on Day 0 (D0), intravitreally treated with mT4 mAb (2 µg/1 µL/eye) on D3 and analyzed for CNV leakage by fluorescein angiography on D7, followed by euthanasia. RPE eyecups were isolated and immunostained for CNV vessel. (**A**) Representative images of fluorescein angiography. (**B**) Quantification of CNV leakage area shown in (**A**). (**C**) Quantification of CNV leakage intensity shown in (**A**). (**D**) Representative images of CNV vessel immunostaining in RPE eyecups. (**E**) Quantification of CNV maximal lesion area shown in (**D**). (**F**) Quantification of CNV 3D volume shown in (**D**). Blind-coded data. n = 19 laser spots/6 eyes (mT4 mAb) and 23 laser spots/6 eyes (PBS). ±SEM; *t*-test. Scale bar = 100 µm.

**Figure 5 ijms-25-09507-f005:**
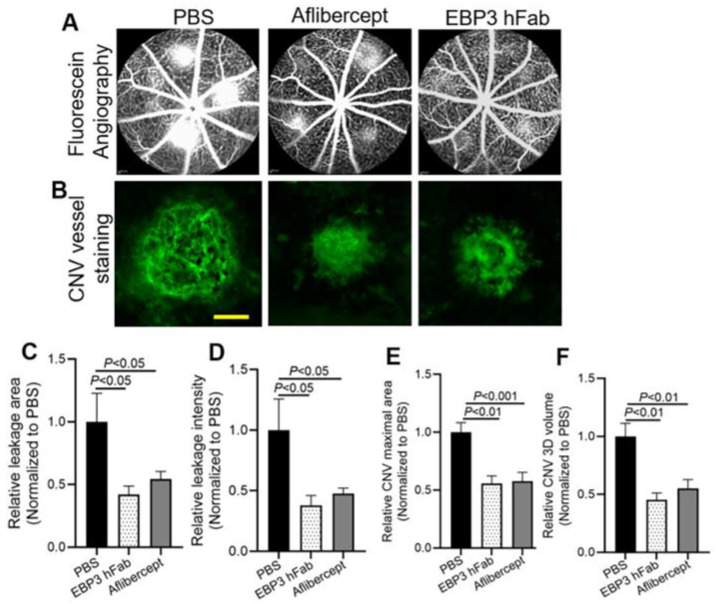
Anti-Scg3 EBP3 hFab ameliorates LCNV in mice. LCNV was induced, treated with EBP3 hFab, aflibercept (2 µg/1 µL/eye) and analyzed, as described in Figure 4. (**A**) Representative images of fluorescein angiography. (**B**) Representative images of immunostained CNV vessels in RPE eyecups. Scale bar = 100 µm. (**C**) Quantification of CNV leakage area in (**A**). (**D**) Quantification of CNV leakage intensity in (**A**). (**E**) Quantification of CNV maximal lesion area in (**B**). (**F**) Quantification of CNV 3D volume in (**B**). Data were blind-coded. n = 21 laser spots (PBS), 20 (EBP3 hFab) and 26 (aflibercept). ±SEM; one-way ANOVA test.

**Figure 6 ijms-25-09507-f006:**
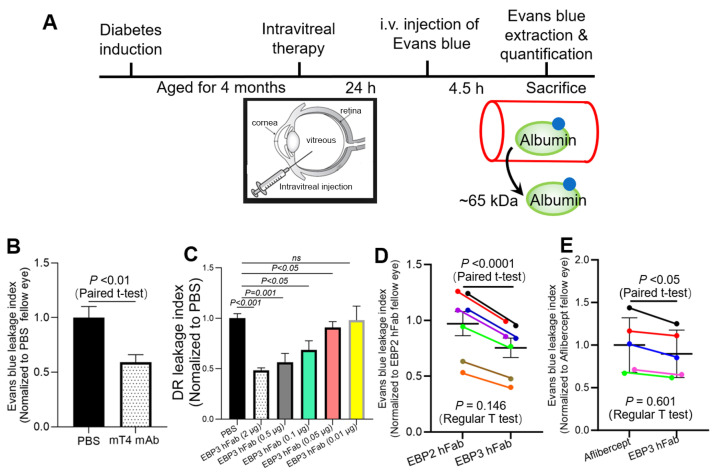
Anti-Scg3 EBP3 hFab alleviates DR leakage more effectively than EBP2 hFab and aflibercept in diabetic mice. (**A**) Illustration of the Evans blue assay to quantify DR leakage. Mice were induced for hyperglycemia and aged for 4 months to develop DR leakage. The indicated therapeutic agents and PBS were intravitreally injected, and Evans blue assay was performed the next day to quantify DR leakage. (**B**) Anti-Scg3 mT4 mAb (2 µg/eye) effectively alleviated DR leakage. (**C**) Dose–response curve of anti-Scg3 EBP3 hFab to alleviate DR leakage in diabetic mice. EBP3 hFab was injected into one eye of the DR mice with PBS for the fellow eye. Only one of five PBS groups is presented as a representative (left-most column). All data are normalized to cognate PBS eyes for paired *t*-test. (**D**) EBP3 hFab is more effective for ameliorating DR leakage than EBP2 hFab. The two hFabs (2 µg/eye) were paired and intravitreally injected into the two eyes of each DR mouse. (**E**) EBP3 hFab is more effective than aflibercept (2 µg/eye), as described in (**D**). Each color line in D and E represents one DR mouse. Data were blind-coded. n = 5 mice/group for all except n = 7 mice/group in (**D**). ±SEM. All data are normalized and compared against the fellow eyes of the same mice by paired *t*-test. Regular two-tailed *t*-test was also performed for (**D**,**E**) with indicated *p* value. Ns—not significant.

**Table 1 ijms-25-09507-t001:** Binding affinity (K_D_) of anti-Scg3 mAbs and hFabs analyzed by Octet.

Antibody	ML49.3 mAb	EBP2 hFab	mT4 mAb	EBP3 hFab
K_D_ (nM)	35	8.7	0.86	0.12

## Data Availability

All data available upon request to corresponding author.

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
