# Peer review of "Optimal Humanized Scg3-Neutralizing Antibodies for Anti-Angiogenic Therapy of Diabetic Retinopathy"

_ijms, 2024, doi:10.3390/ijms25179507_

Round 1
Reviewer 1 Report
Comments and Suggestions for Authors
The manuscript entitled Optimal humanized Scg3-neutralizing antibodies for anti-angiogenic therapy of diabetic retinopathy is an original article. The authors aimed to identify the preferred humanized antibody Fab fragment (hFab) for diabetic retinopathy therapy; therefore, they compared all four antibodies (ML49.3 mAb, EBP2 hFab, mT4 mAb, and EBP3 hFab) for binding neutralizing and therapeutic activities in vitro and in vivo. The hFabs comparing with mAbs improved binding affinities. They identified EBP3 hFab as the preferred hFab to develop anti-Scg3 therapy. From a practical point a view this study is important for preclinical anti-angiogenic drug screening to select the best candidates for clinical translation.
The manuscript is very well written. The results are well presented. Discussions are short. However, I have some suggestions.
I recommend extending the discussions with the response for the following questions. However, I wondering what are the obstacles for moving to phase 1 studies with this anti-angiogenic agent? Could you emphasize the steps towards the phase 1 study? What are the gaps through this?
Author Response
- I recommend extending the discussions with the response for the following questions. However, I wondering what are the obstacles for moving to phase 1 studies with this anti-angiogenic agent? Could you emphasize the steps towards the phase 1 study? What are the gaps through this?
Response: Anti-Scg3 hFab is currently under development for translation. Major obstacles to Phase I clinical trials are Investigational New Drug (IND)-enabling studies, including Good Manufacturing Practice (GMP) manufacturing of the hFab, Good Laboratory Practice (GLP) toxicology and pharmacokinetics. Once these studies are completed, an IND application will be submitted to the U.S. Food and Drug Administration (FDA).
We added the following sentence at the end of Discussion: “Based this work, EBP3 hFab has been selected for Investigational New Drug (IND)-enabling preclinical studies to develop the biologic toward clinical trials.” (Line 390-392)

Reviewer 2 Report
Comments and Suggestions for Authors
In my point of view some revisions are needed before the manuscript submitted by Huang et al. can be considered for publication in IJMS.
Here are my suggestions:
Line 18: “in vitro” and “in vivo” should be in italics. Please, revise it in the whole manuscript.
Please, specify clearly, your main results in the abstract and some directions for future studies.
Line 31: Explain the meaning of “FDA”.
The Introduction section needs to be expanded. A better justification for the need for the present study has to be given. A deeper background is also recommended. At the end of this section is expected the authors can provide clearly the objectives of the study.
“The results suggest that anti-Scg3 EBP3 hFab is an optimal candidate to be developed for clinical translation. The in vivo efficacy comparison assay in the DR model is also valuable to reliably evaluate efficacy of any two anti-angiogenic biologics.” – This shouldn’t be placed at the end of the introductory section. Here, as I previously mentioned, you should state the main goals of your work.
Lines 311-325: References are missing in this part of the Discussion section.
I miss a strengths and limitations section as well as a Conclusions section. Here, in conclusion, you should also provide future perspectives.
Author Response
Reviewer #2
- Line 18: “in vitro” and “in vivo” should be in italics. Please, revise it in the whole manuscript.
Response: Revised as suggested. In vitro and in vivo in other sections are also now in italic.
- Please, specify clearly, your main results in the abstract and some directions for future studies.
Response: The main results are summarized in the last sentence in the Abstract section. directions for future studies are described in Discussion (Line 380-382 & 390-392).
- Line 31: Explain the meaning of “FDA”.
Response: In Line 31, we revised “The FDA approval of anti-angiogenic drugs against vascular endothelial growth factor (VEGF) represents….” to “Anti-angiogenic drugs against vascular endothelial growth factor (VEGF) approved by the U.S. Food and Drug Administration (FDA) represent….”
- The Introduction section needs to be expanded. A better justification for the need for the present study has to be given. A deeper background is also recommended. At the end of this section is expected the authors can provide clearly the objectives of the study.
Response: We included additional background. Objectives were added to the end of the Introduction section.
- “The results suggest that anti-Scg3 EBP3 hFab is an optimal candidate to be developed for clinical translation. The in vivo efficacy comparison assay in the DR model is also valuable to reliably evaluate efficacy of any two anti-angiogenic biologics.” – This shouldn’t be placed at the end of the introductory section. Here, as I previously mentioned, you should state the main goals of your work.
Response: We removed these two sentences and added objectives at the end of the Introduction section.
- Lines 311-325 (now Line 338-352): References are missing in this part of the Discussion section.
Response: This section of Discussion recapitulates the unique experimental features and results in Figure 6D and 6E. Variation in DR severity among diabetic mice is evidenced by the error bars in Figure 6D and 6E and P values between paired and two-tailed t-tests. The shared systemic hyperglycemia for the left and right eyes of each mouse is self-evidenced. Equal degree of intracardial perfusion for two eyes of the same mice is also a common knowledge. As a result, inter-mouse variations in efficacy are anticipated to be greater than intra-mouse variations. No reference is needed for this paragraph.
- I miss a strengths and limitations section as well as a Conclusions section. Here, in conclusion, you should also provide future perspectives.
Response: Strengths of this study are summarized in Conclusion. The strength of anti-Scg3 therapy for disease-targeted therapy are discussed throughout the manuscript (e.g., Line 298-300). A major limitation of anti-Scg3 is the lack of clinical trial data at this stage (Line 382). Strengths and limitations of the DR models for paired efficacy analysis are elaborated in Discussion (Line 338-352 and 366-377, respectively). A new Conclusion section is added.

Round 2
Reviewer 2 Report
Comments and Suggestions for Authors
The manusdrip can be accepted after these revisions and improvements made by the authors.